# Standardization of Post-Vitrification Human Blastocyst Expansion as a Tool for Implantation Prediction

**DOI:** 10.3390/jcm11092673

**Published:** 2022-05-09

**Authors:** Anat Hershko-Klement, Shaul Raviv, Luba Nemerovsky, Tal Rom, Ayelet Itskovich, Danit Bakhshi, Adrian Shulman, Yehudith Ghetler

**Affiliations:** 1The IVF Unit, Department of Obstetrics and Gynecology, Hadassah Mt Scopus, The Hebrew University of Jerusalem, Jerusalem 91240, Israel; 2The IVF Unit, Meir Medical Center, Department of Obstetrics and Gynecology, Affiliated to Sackler School of Medicine, Tel Aviv University, Kfar Saba 44281, Israel; shauli.raviv@gmail.com (S.R.); lnemerovsky@gmail.com (L.N.); tal4rom@gmail.com (T.R.); ayelet.itskovich@clalit.org.il (A.I.); danit.bakhshi@gmail.com (D.B.); adrian@adrianshulman.co.il (A.S.); yghetler@gmail.com (Y.G.)

**Keywords:** vitrification, blastocyst, expansion, pregnancy rate, in vitro fertilization

## Abstract

The increased use of vitrified blastocysts has encouraged the development of various criteria for selecting the embryo most likely to implant. Post-thaw assessment methods and timetables vary among investigators. We investigated the predictive value of well-defined measurements of human blastocyst re-expansion, following a fixed incubation period. Post-thaw measurements were taken exactly at 0 and 120 ± 15 min. Minimum and maximum cross-sectional axes were measured. Three groups were defined: Group 1: embryos that continued to shrink by 10 µm or more; group 2: embryos that ranged from −9 to +9 µm; and group 3: re-expansion of 10 µm or more. Patient and morphokinetic data were collected and integrated into the analysis. A total of 115 cases were included. The clinical pregnancy rate for group 1 was 18.9%; group 2, 27%; and group 3, 51.2% (*p* = 0.007). Pre-thaw morphologic grading and morphokinetic scores of the study groups did not reveal differences. *p*-values were 0.17 for the pre-thaw morphologic score, 0.54 for KID3, and 0.37 for KID5. The patients’ demographic and clinical data were similar. The clinical pregnancy rate correlated with the degree of thawed blastocyst re-expansion measured 2 h after incubation. This standardized measure is suggested as a tool to predict the potential of treatment success before embryo transfer.

## 1. Introduction

Time-lapse monitoring, improved culture systems and vitrification, have enabled blastocysts to be used successfully in IVF treatment programs [1,2]. Vitrification has improved the strategy of elective single-embryo transfer for fresh cycles, enabling efficient freezing of surplus embryos and a “freeze all” approach for preventing ovarian hyperstimulation syndrome. It has also enhanced preimplantation genetic testing. These factors have led to the establishment of treatment cycles involving frozen blastocysts [3].

The increased use of vitrified blastocysts has encouraged the development of more accurate selection criteria. Traditionally, the Gardner grading system [4] for fresh blastocysts was widely used in ART to select embryos. It is based on morphological appearance and involves three parameters: degree of blastocoele expansion, trophectoderm (TE), and inner cell mass (ICM). Additional information can be obtained from morphokinetic evaluation of the blastocyst cultured in a time-lapse incubator [5]. However, after the vitrification and warming procedures, blastocysts undergo several morphological changes that may make it difficult to evaluate their quality. First, they are dehydrated by adding freezing solutions during cooling and then rehydrated by removing thawing solutions during warming. Blastocoele shrinking and swelling can damage cells and may affect morphological integrity and survival [6,7]. Blastocysts often are collapsed immediately after warming and the ability to assess their ICM and TE is very limited. In addition, expanded human blastocysts will contract in response to any major physical or chemical change in their surroundings, such as decreases in temperature, changes in culture media or during aspiration into a pipette [8]. The formation and maintenance of the blastocyst cavity (blastocoel) are attributed to the sodium pump (Na+/K+−ATPase) [9], as well as the creation of aquaporins across the epithelium into the extracellular space of the blastocyst to form the fluid-filled blastocoel [10]. Therefore, the ability to re-expand may imply the proper function of the blastocyst. Thus, a culture period after warming provides the opportunity to evaluate vitrified-warmed blastocysts more accurately.

Some investigators recommend assessing survival and quality within 2–4 h after warming [6,7]. The ability to re-expand within a few hours after warming has been reported to be a strong indicator of blastocyst potential [11]. However, assessment methods and timetables vary among investigators. Post-thaw re-expansion was measured in time-lapse systems by Maezawa et al. [12] at 5–6 h, at 3.5–7 h by Coello et al. [13], for up to 5.67 h by Kovacic et al. [14], for a range of 20 min to 4 h and 42 min by Giunco et al. [15] and for up to 22 h by Iwasawa et al. [16]. Others reported gross microscopic evaluation by embryologists 1–6 h post-thaw [17] or 2 h post-thaw [18,19]. In addition, the comparison was sometimes performed on blastocysts that underwent artificial shrinkage pre-vitrification [20].

The current study investigated the predictive value of well-defined, quantitative measurements of re-expansion of frozen-thawed blastocysts, following a defined period of incubation.

## 2. Materials and Methods

This retrospective cohort study included vitrified-thawed blastocyst transfers carried out from November 2017 to January 2020, in a hospital-based IVF unit. In our lab, frozen-thawed blastocysts are observed immediately after warming and again prior to transfer. Pictures are taken for documentation and blastocysts are measured at ×300 magnification.

For standardization, blastocyst transfers were included in the study only when the pre-vitrification developmental morphokinetic data were documented; this is, when two clear post-thaw microscopic images (×300 magnification) were available, if the first was performed immediately following the thawing process and the second after 120 ± 15 min of culturing in an incubator, and when the implantation data (KID) were documented. KID score day 3 assigns a morphokinetic score from 1 to 5 to annotated embryos. The score from 1 to 5 is a relative measure of the implantation potential of the embryo and is based on timing annotations. KID day 5 scoring considers the morphology and the morphokinetic traits of an embryo. For each embryo, the model calculates a continuous score from 1 to 9.9. The score reflects the statistical chance of implantation based on development information from the five-day culture period. The higher the score, the greater the statistical chance of implantation. Our study protocol inclusion criteria included strict measurements at time 0 and time 120 ± 15 min post-thaw. In addition, for transfer eligibility, at least 50% of the cells were required to be intact. During the study period, 324 blastocysts were thawed and transferred after various times, ranging from 10 min to 4 h of incubation. We included in the study only blastocysts that met the inclusion criterion of 120 ± 15 min of incubation, since this period was previously reported as a point of re-expansion assessment and could be followed in our lab, if proven valuable.

Minimum and maximum cross-section axes of the blastocysts were measured and the average was calculated. Each line in the measurement scale represents 10 µm (Figure 1). The difference in size between the average of time 0 measurements (immediately after thawing) and time 2 (two hours after thawing) was calculated.

For analysis, the transferred blastocysts were divided into 3 groups according to their re-expansion pattern. The patterns were calculated by subtracting the diameter immediately post-thaw from the 2 h post-thaw diameter. Group 1 included embryos that continued to shrink by 10 µm or more after the 2 h measurement; group 2, embryos with measurement differences ranging from −9 to 9 µm according to the 2 h measurement; and group 3 included those demonstrating a re-expansion trend of 10 µm or more.

Patient data regarding age at egg retrieval, treatment indication, cycle protocol, number of oocytes, endometrial lining, morphokinetic events and embryo grading prior to vitrification were collected, as well as the clinical pregnancy rate, confirmed by documentation of a gestational sac, as detailed below.

### 2.1. Embryo Vitrification Protocol and Embryonic Data

All blastocysts were cultured in Global culture media (LifeGlobal, Brussels, Belgium) in a time-lapse incubator (Vitrolife, Gothenburg, Sweden). Surplus blastocysts were vitrified. Detailed morphokinetic variables, including uneven blastomere, multinucleation, direct division, irregular division, rapid division, time for each division, start of compaction, start of blastulation and expansion, and pumping, as well as Gardner score, were evaluated by the same experienced senior embryologist (Y.G.)

KID scores were calculated from the time-lapse data. KID3 (score on day 3) considered pronuclei (PN) using time points for pronuclei fading (PNf), t2, t3, t5 and t8 [21]. KID5 (score on day 5), considered pronuclei (PN) using time points for t2, t3, t4, t5, t8 and TE grading [22,23].

Freezing and thawing were performed using the SAGE kit (Cooper Surgical, Trumbull, CT, USA), according to the manufacturer’s recommendation. The freezing device used was either a Cryotop (Kitazato, Japan) or Cryolock (Biotec, Alpharetta, GA, USA). One blastocyst was loaded per device.

### 2.2. Endometrial Preparation Protocol

Patients were prepared for the transfer with one of the following 3 protocols: 1. Hormonal replacement protocol using 6 mg micronized estradiol divided into 3 doses daily (2 mg estrofem, Rottapharm, Italy) starting on day 2 of the cycle, for 8–10 days until endometrial thickness was > 7 mm. Once this thickness was measured by trans-vaginal ultrasound, 300 mg natural vaginal progesterone divided into three 100 mg doses (Endometrin, Floris LTD, Israel) was added for 5 days before transfer. 2. Natural cycle protocol with a transfer performed on day 7 post-LH surge, or 3. Ovulation induction with 2.5 mg aromatase inhibitor (Femara, Novartis, Israel) starting on day 3 of a natural or artificially induced cycle, for a total of 5 days. Ovulation was triggered using 250 µg recombinant hCG (Ovitrelle, Merck Serono, Italy) once a leading follicle and >7 mm endometrial thickness were reached. Transfer was performed 7 days after triggering.

Clinical pregnancy was defined as the presence of an intrauterine gestational sac detected by trans-vaginal ultrasound at 6 weeks of pregnancy and was the primary outcome measure.

### 2.3. Statistical Analysis

All analyses were performed using SPSS 23.0 (IBM Corp., Armonk, NY, USA). Chi-square or Fisher’s exact test was used to compare rates and proportions. Student *t*-test was used to analyze continuous variables. All *p*-values were tested as two-tailed and considered significant at <0.05.

## 3. Results

During the study period, 115 thawed blastocysts from 115 patients met the selection criteria: group 1 included 37 embryos (32.2%), group 2 included 37 (32.2%) embryos and group 3 included 41 (35.7%) embryos. Examples of the measurements and group criteria are represented in Figure 2.

Values of the changes in diameter for the entire cohort are depicted in Figure 3.

Pre-thaw morphologic expansion grading and morphokinetic scores (Table 1), as well as demographic and clinical data were not significantly different (Table 2 and Table A1) among the groups. Importantly, the three endometrial preparation methods (hormone replacement, natural cycles and letrozole protocols) were distributed similarly among the three expansion groups: group 1 (67.6%, 21.6% and 10.8%); group 2 (54.1%, 21.6% and 24.3%) and group 3 (41.5%, 31.7% and 26.8%), respectively.

The re-expansion patterns of frozen-thawed blastocysts were strongly associated with pregnancy rates (Table 2). While group 1 had only an 18.9% clinical pregnancy rate, the re-expanding embryos (Group 3) reached a 51.2% clinical pregnancy rate (*p* = 0.007). Importantly, the distribution of clinical pregnancy was not significantly different across the endometrial lining preparation protocols (*p* = 0.45).

Correlations between re-expansion and clinical pregnancy according to patient age (<35 or ≥35 years) were evaluated and the same trends were found regardless of age (Table 3). Interestingly, the pregnancy rate for group 3 embryos was about 50% in both age groups.

## 4. Discussion

This study found a strong association between the magnitude of post-vitrification blastocyst re-expansion and pregnancy rates. This finding was maintained in patients over 35 years of age. We used strict evaluation criteria, based on measurements at precisely 2 h post-thaw to evaluate changes in blastocyst size. All other parameters analyzed including demographic data, infertility factors, morphometric parameters, and embryo quality, as expressed by KID score data, were comparable between groups. This indicates that the amount of re-expansion in thawed blastocysts can be reliably used to assess implantation potential.

Several investigators estimated the re-expansion of frozen–thawed blastocysts as a marker of viability and implantation potential. Coello et al. [13] reported that re-expansion was strongly predictive of pregnancy outcomes: 44.6% clinical pregnancy rate in re-expanded embryos vs. 6.5% in those that did not. Based on morphological grade, Du et al. [18] found that the degree of re-expansion was independently correlated with the chances of live birth. Lin et al. [24] reported that the speed of re-expansion was a significant indicator (*p* < 0.01): re-expansion in less than 1 h yielded a 70% pregnancy rate, 1–2 h: 51.8%, and more than 2 h: 28.7%. In their study, the quick re-expansion group included significantly more top-quality embryos compared to the other two groups. Similarly, Ebner et al. [25] reported that the time to achieve re-expansion was significantly shorter in cycles that resulted in pregnancy (*p* < 0.05). Ozgur et al. [26] demonstrated a relationship between decreased expansion and reduced implantation rates. However, Giunco et al. [15] did not find that the re-expansion status of blastocysts is indicative of their implantation potential; although, the time span for assessment was sometimes as short as 20 min.

Comparing survival rates and morphological parameters of thawed blastocysts in the literature is challenging, because some embryologists focused on immediate survival, while others suggested waiting up to 24 h to monitor survival and growth [14,16,19]. To date, most reported data are based on a visual assessment by one or two experienced embryologists [19,24]. Visual morphological evaluation of embryos can be very subjective [27]. In addition, some reports monitored re-expansion with a time-lapse system, but the timing of these measurements varied considerably. Kovacic et al. [14] used 2.33 h, Ebner et al. [25] 4.6 ± 1.2 h, Maezawa et al. [12] 5–6 h, Coello et al. [13] 3.5–7 h and Iwasawa et al. [16] up to 22 h. Furthermore, different approaches were described regarding handling the blastocysts, such as collapsing before freezing [14,18] or post-thaw artificial hatching [7,18,25]. These variations limit the ability of IVF facilities to adopt a single method for evaluating re-expansion.

Accurate scoring of vitrified and warmed day 5 embryos allows adequate prediction of treatment outcomes. However, at the end of the thawing procedure, the vitrified blastocysts usually shrink and the ability to assess their ICM and TE is limited. It is difficult to apply the conventional grading system in these conditions. An additional tool that can identify the ability to implant and is convenient for busy IVF units is needed.

The blastocyst measurements provide accurate data about re-expansion, and the two-hour culture period after thawing used in this study seems suitable for the workflow. Therefore, objective measures of expansion and the resulting clinical pregnancy rate from thawed blastocysts provide feasible, predictive variables and have great value for evaluating the chances of treatment success prior to embryo transfer.

The current study was based in a single lab with a well-defined measurement protocol and timetable. We also included the time-lapse parameters before freezing in the analysis. The study is limited by its retrospective nature, small sample size and use of different endometrial preparation protocols. However, the preparation protocols reflect real-life protocols and were distributed evenly among the three study groups, all of which had sufficient endometrial thickness. All other demographic and clinical data were similar among the study groups. The single parameter that demonstrated a statistical difference was the clinical pregnancy rate attributed to the re-expansion of the post-thaw blastocysts.

The current study may assist in decisions regarding embryo transfer protocols among patients with embryos that do not re-expand. In these cases, we may consider thawing an additional embryo (when available) or discuss future changes in vitrification timing, such as the day of freezing (day 3 instead of day 5) or possibly modifying the lab protocol (e.g., blastocyst incubation time in equilibration media or media used). Therefore, we suggest using the degree of post-thaw re-expansion at 2 h as an additional parameter for evaluating blastocyst implantation potential in vitrified–warmed cycles.

## 5. Conclusions

The degree of thawed blastocyst re-expansion, measured 2 h after post-thaw incubation, correlates with clinical pregnancy rates. This objective measure may serve as a predictive tool for the potential of treatment success before embryo transfer.

## Figures and Tables

**Figure 1 jcm-11-02673-f001:**
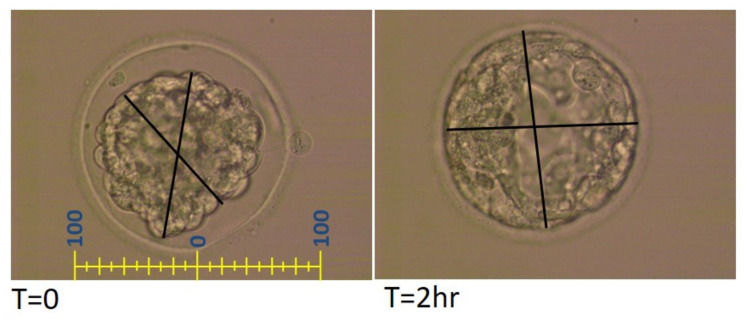
Measurement protocol of thawed blastocysts.

**Figure 2 jcm-11-02673-f002:**
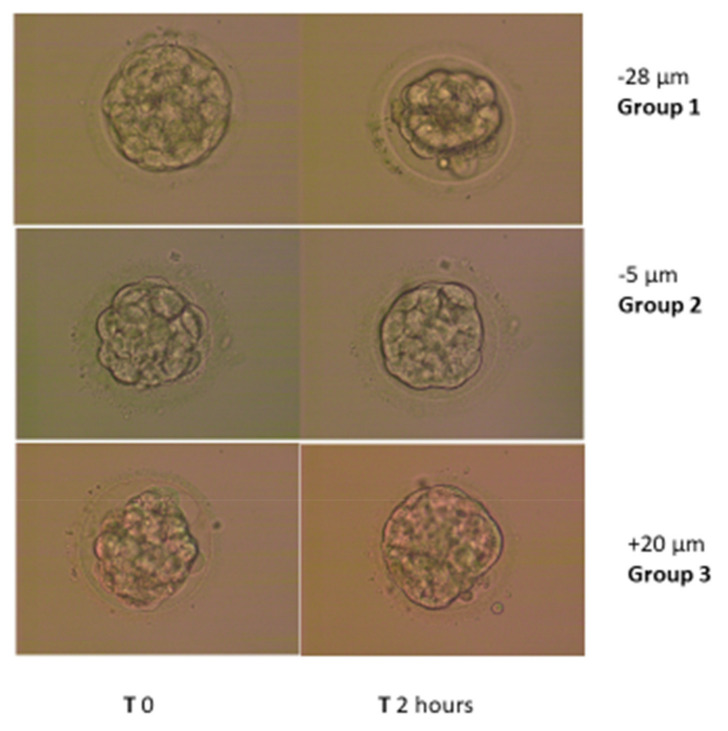
Examples of embryonic measurements and grouping criteria.

**Figure 3 jcm-11-02673-f003:**
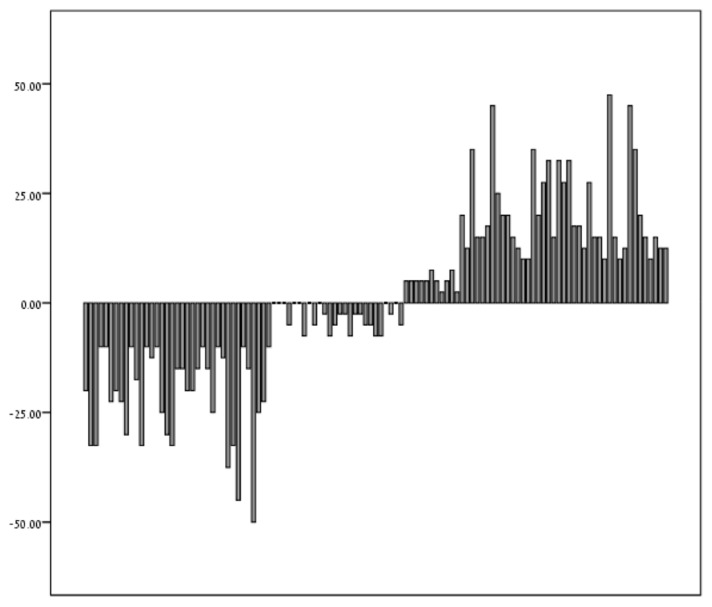
Histogram illustrating changes in size between the two-hour diameter and immediate post-thaw diameter for each of 115 embryos (µm).

**Table 1 jcm-11-02673-t001:** Pre-vitrification embryonic data as expressed by expansion score and KID score.

Study Group * (*n*)	Pre-Freeze Gardner’s Expansion Score				KID Score Day 3	KID Score Day 5
	2 *n* (%)	3 *n* (%)	4 *n* (%)	5 *n* (%)		
1 (37)	5 (13.5)	7 (18.9)	24 (64.9)	1 (2.7)	4.27 ± 1.26	6.92 ± 1.35
2 (37)	8 (21.6)	10 (27.0)	17 (45.9)	2 (5.4)	4.27 ± 1.17	6.48 ± 1.79
3 (41)	2 (4.9)	8 (19.5)	26 (63.4)	5 (12.2)	4.51 ± 0.87	6.94 ± 1.62
*p*-value	0.17				0.54	0.37

Numbers represent mean ± SD or *n* (%); * Group 1—Shrunk ≥10 µm; group 2—Change between −9 and +9 µm; group 3—re-expansion trend ≥10 µm.

**Table 2 jcm-11-02673-t002:** Main demographic, clinical and clinical pregnancy data according to study groups.

Study Group* (*n*)	Age (at Oocyte Retrieval)	N Oocytes Aspirated	Endometrial Lining, mm	Fertilization Method (%)		Clinical Pregnancy
				IVF	ICSI	
1 (37)	33.65 ± 6.01	12.38 ± 6.08	8.6 ± 2.1	54.1	45.9	7 (18.9%)
2 (37)	32.89 ± 5.04	13.89 ± 7.12	9.4 ± 1.9	67.6	32.4	10 (27.0%)
3 (41)	33.63 ± 6.07	14.22 ± 8.31	9.1 ± 1.5	57.9	42.1	21 (51.2%)
*p*-value	0.88	0.50	0.16	0.49		0.007

Numbers represent mean ± SD or *n* (%); * Group 1—Shrinking ≥10 µm; group 2—Change between −9 and +9 to µm; group 3—re-expansion trend ≥10 µm.

**Table 3 jcm-11-02673-t003:** Clinical pregnancy rate according to age at oocyte retrieval.

Study Group *	Age < 35 Years	Age ≥ 35 Years
	Clinical Pregnancy *n* (%)	Clinical Pregnancy *n* (%)
Group 1	20 (25)	17 (11.8)
Group 2	23 (34.8)	14 (14.3)
Group 3	21 (52.4)	20 (50.0)
*p*-value	0.07	0.009

* Group 1—Shrunk ≥10 µm; group 2—Change between −9 and +9 µm; group 3—re-expansion trend ≥10 µm.

## Data Availability

Upon request.

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
