# Peer review of "Standardization of Post-Vitrification Human Blastocyst Expansion as a Tool for Implantation Prediction"

_jcm, 2022, doi:10.3390/jcm11092673_

Round 1

Reviewer 1 Report

In this study the authors aims to investigate the predictive value of blastocysts re-expansion soon after vitrification and thawing and 2 hours post- thawing.

The idea is interesting but no new. The material and method section is not clear especially in the description of the 3 experimental groups. Why did you choose these intervals of meausurements? 

2.1 define KID scores

Some imprecisions  all over the manuscript. Please be concordant in your text: vitrification is different from cryopreservation.

Some revison of the english is required especially in the introduction.

Author Response

Reviewer 1

# In this study the authors aims to investigate the predictive value of blastocysts re-expansion soon after vitrification and thawing and 2 hours post- thawing.

The idea is interesting but no new. The material and method section is not clear especially in the description of the 3 experimental groups.

Thank you for your positive comment. The material and methods section was revised, especially the description of the 3 study groups (lines 100-106 in the revised clean version).

#Why did you choose these intervals of measurements? 

This specific 10 micron interval was chosen because it is the minimum discriminatory value in our measurement tool.

# Define KID scores

KIDScore D3 assigns a morphokinetic score from 1-5 to annotated embryos. The score from 1-5 is a relative measure of the implantation potential of the embryo and is based on annotations of: PN assessment, PN fading, time (t) to 2, 3, 4, 5 and 8 cells.

KIDScore D5 considers the morphology and the morphokinetic traits of an embryo. For each embryo, the model calculates a continuous score from 1-9.9. The score reflects the statistical chance of implantation based on development information from the 5-day culture period. The higher the score, the greater the statistical chance of implantation.

This explanation was added to the manuscript, Materials and methods (lines 80-83 in the revised clean version)

# Some imprecisions  all over the manuscript. Please be concordant in your text: vitrification is different from cryopreservation.

Thank you. The paper was revised by an English-language editor and imprecisions corrected .

Some revison of the english is required especially in the introduction.

Thank you. The paper was revised by an English-language editor.

Reviewer 2 Report

The authors of this retrospective study have examined the effect of the degree of expansion of previously vitrified blastocyst embryos on the outcome of embryo transfer after thawing. As the authors correctly point out in the introduction, the are currently no unified protocols dealing with the timing of thawing of blastocyst and the duration of their culture until embryo transfer. Many clinical embryology laboratories indeed prefer to observe the expansion of thawed blastocyst embryos before proceeding to embryo transfer. Several reports have demonstrated that blastocoel expansion is a favorable prognostic sign. In this retrospective study, the authors have now used a fixed protocol, in which embryos were photographed immediately after thawing and again after two hours. In addition, the size of the blastocysts (“expansion”) was measured at each of both instances and based on the objective degree of expansion the embryos were divided into three groups: 1. shrinkage of at least 10 μm; 2. Intermediate groups with either shrinkage less than 9 μm or expansion of less than 9 μm; 3. expansion of 10 μm or more. After having corrected for a number of confounding variables (including pre-vitrification morphology, as given by time lapse) the authors concluded that group 3 was the most promising. This finding fits well to published literature.

The manuscript is well written and both the analysis and the presentation of the results are straight-forward. The number of embryos in each of the three groups is similar.

There are, however, some questions that arise while reading the manuscript (repeatedly):

  1. Whereas the authors label their study as “retrospective”, the fixed and uniform protocol of analyzing the thawed blastocysts and to image their development after thawing twice rather suggest a prospective study protocol. Ethical approval is mentioned at the end of the manuscript. The authors should explain, how a uniform and fixed assessment of thawed blastocyst embryos was installed and why this study is still labeled as being retrospective?
  2. Three different protocols for preparation of the endometrium prior to thawing and embryo transfer are given, but it was not demonstrated, whether the choice of any of these three protocols impacted on the pregnancy rate. The choice of the endometrial preparation may have been a confounding factor.
  3. One of the three protocols consisted of 6 mg micronized estradiol daily (taken orally) and 300 mg of vaginal progesterone (Endometrin). The latter is an unusually low daily dose and might contribute to higher (early) miscarriage rates. The authors should verify this dosage and add miscarriage rates to their list of outcome parameters.
  4. The authors have not detailed their embryo transfer policy? How many embryos were thawed during each thawing cycle and how many embryos were transferred per treatment cycle? If more than one embryo was transferred per treatment trial, then how were the authors able to connect a pregnancy with the morphology of the embryos?
  5. Table 2: In addition to the number of aspirated oocytes, some additional information on the number of diploidic fertilized oocytes and on the number of embryos that developed up to the blastocyst stage would be helpful to appreciate any potential differences in the three groups.
  6. An additional Figure with a representative example of blastocysts as included in each of the three groups would be nice.
  7. How did the authors decide to group the embryos according to the proposed cutoff limits given in their retrospective study? The authors might add a figure with a histogram or distribution of all measurements of expansion, from which the interested reader may well deduce, why the cutoffs were chosen for the grouping of the thawed blastocysts.
  8. Finally, the authors may add to the discussion some background information about the known physiology of blastocyst expansion, including the energy-dependent ion pump which expands the blastocoel of good quality embryos.
  9. The reference list seems rather incomplete. The most recent citation dealing with blastocyst expansion after thawing dates from 2020 and the second most recent citation dates from 2019. The authors might update their reference list.

Minor issues:

Table 2: instead of “insemination” I suggest to use “IVF”, which stands in better contrast to ICSI.

The compound estradiol (6 mg daily) is not natural estradiol, but micronized estradiol.

Author Response

# The manuscript is well written and both the analysis and the presentation of the results are straight-forward. The number of embryos in each of the three groups is similar.

We thank the reviewer for this comment.

There are, however, some questions that arise while reading the manuscript (repeatedly):

# Whereas the authors label their study as “retrospective”, the fixed and uniform protocol of analyzing the thawed blastocysts and to image their development after thawing twice rather suggest a prospective study protocol. Ethical approval is mentioned at the end of the manuscript. The authors should explain, how a uniform and fixed assessment of thawed blastocyst embryos was installed and why this study is still labeled as being retrospective?

All thawed blastocysts are documented in our lab by photography taken at time 0' and time of transfer. For the purpose of the study, we explored a uniform incubation time and assessment. During the study period, 324 blastocysts were thawed and transferred after various times, ranging from 10 minutes to 4 hours of incubation. We included in the study only blastocysts that met the inclusion criterion of 120 ± 15 minutes of incubation, since this period was previously reported as a point of re-expansion assessment and could be followed in our lab, if proven valuable. This information was clarified in the manuscript (lines 88-93 in the revised clean version).

# Three different protocols for preparation of the endometrium prior to thawing and embryo transfer are given, but it was not demonstrated, whether the choice of any of these three protocols impacted on the pregnancy rate.

We added the analysis of clinical pregnancy according to FET protocol (lines 170-171 in the revised clean version). The distribution of clinical pregnancy was not significantly different across endometrial lining preparation protocols (P=0.45).

# The choice of the endometrial preparation may have been a confounding factor.

One of the three protocols consisted of 6 mg micronized estradiol daily (taken orally) and 300 mg of vaginal progesterone (Endometrin). The latter is an unusually low daily dose and might contribute to higher (early) miscarriage rates. The authors should verify this dosage and add miscarriage rates to their list of outcome parameters.
We thank the reviewer for this observation.  The study assessed clinical pregnancies – those presenting a compatible gestational sac by 6 weeks. In our practice, at the point of 6 gestational weeks, patients are routinely discharged from the IVF unit to the community, for pregnancy follow-up. Therefore, we do not have information regarding future miscarriages. The  6mg oral Estradiol and 300mg daily dose of vaginal Endometrin is successfully practiced in our unit and was detailed as an FET protocol by Anat Hershko Klement (first author) et al. Does fresh single embryo transfer outcome predict the result of a subsequent vitrified-warmed blastocyst of the same cohort? Hum Fertil (Camb) 2020 Jul 20;1-6. doi: 10.1080/14647273.2020.1794061.

# The authors have not detailed their embryo transfer policy? How many embryos were thawed during each thawing cycle and how many embryos were transferred per treatment cycle? If more than one embryo was transferred per treatment trial, then how were the authors able to connect a pregnancy with the morphology of the embryos?

Our preferred policy for thawed blastocysts is single embryo transfer. All embryos in the study were frozen-thawed day 5 embryos.  However, when 2 embryos were transferred, we included only cases in which the fate of both embryos was known (both implanted or failed to implant).

# Table 2: In addition to the number of aspirated oocytes, some additional information on the number of diploidic fertilized oocytes and on the number of embryos that developed up to the blastocyst stage would be helpful to appreciate any potential differences in the three groups.
Our database was based on a query that did not encompass information regarding diploidic fertilized oocytes and blastulation of the cohort from which the embryo originated. Therefore, we cannot provide this information.  However, even if the shrinking blastocyst originated from a cohort characterized by lower fertilization and blastulation rates, the specific embryo included was still of high quality, making it eligible for vitrification and eventual transfer.

# An additional Figure with a representative example of blastocysts as included in each of the three groups would be nice.

Please find this additional imaging. It was incorporated into the manuscript as Figure 2.

# How did the authors decide to group the embryos according to the proposed cutoff limits given in their retrospective study? The authors might add a figure with a histogram or distribution of all measurements of expansion, from which the interested reader may well deduce, why the cutoffs were chosen for the grouping of the thawed blastocysts.

The specific 10 micron interval was chosen because it is the minimum discriminatory value in our measurement tool.

Thank you for your suggestion to add a histogram. It was added to the manuscript as Figure 3.

# Finally, the authors may add to the discussion some background information about the known physiology of blastocyst expansion, including the energy-dependent ion pump which expands the blastocoel of good quality embryos.

The reference list seems rather incomplete. The most recent citation dealing with blastocyst expansion after thawing dates from 2020 and the second most recent citation dates from 2019. The authors might update their reference list.

 Thank you for this important suggestion. We added information regarding the physiology of expansion (lines 48-53 in the revised clean version) and updated the reference list, adding references 9, 10, 18 and 26.

# Minor issues:

Table 2: instead of “insemination” I suggest to use “IVF”, which stands in better contrast to ICSI.

As suggested, “insemination” was changed to “IVF” in table 2.

The compound estradiol (6 mg daily) is not natural estradiol, but micronized estradiol.

Thank you.– This was revised in the Methods section (line 128 in the revised clean version).

please see the atachment

Round 2

Reviewer 2 Report

The manuscript has benefited from the changes introduced by the authors.